

# Hydrometeorological Data from a Remotely Operated Multi-Parameter Station network in Central Asia

Cornelia Zech (1), Tilo Schöne (1), Julia Illigner (1), Nico Stolarczuk (1), Torsten Queißer (1), Matthias Köppl (1), Heiko Thoss (1), Alexander Zubovich (2), Azamat Sharshebaev (2), Kakhramon Zakhidov (3), Khurshid Toshpulatov (3), Yusufjon Tillayev (4), Suhrob Olimov (5), Zabihullah Paiman (6), Katy Unger-Shayesteh (1), Abror Gafurov (1), Bolot Moldobekov (2)

(1) Helmholtz Centre Potsdam GFZ German Research Centre for Geosciences, Potsdam, Germany
(2) Central-Asian Institute for Applied Geosciences (CAIAG), Bishkek, Kyrgyzstan
(3) Centre of Hydrometeorological Service (UzHydromet), Tashkent, Uzbekistan
(4) Ulugh Beg Astronomical Institute (UBAI) of the Uzbek Academy of Sciences , Tashkent, Uzbekistan
(5) Agency for Hydrometeorology, Committee on Environmental Protection under the Government of the Republic of Tajikistan, Dushanbe, Tajikistan
(6) Kabul Polytechnic University, Kabul, Afghanistan

## Abstract

The Regional Research Network „Water in Central Asia" (CAWa) funded by the German Federal Foreign Office consists of 18 remotely operated multi-parameter stations (ROMPS) in Central Asia. These stations were installed by the German Research Centre for Geosciences (GFZ) in Potsdam, Germany in close cooperation with the Central-Asian Institute for Applied Geosciences (CAIAG) in Bishkek, Kyrgyzstan, the national hydrometeorological services in Uzbekistan and Tajikistan, the Ulugh Beg Astronomical Institute in Tashkent, Uzbekistan, and the Kabul Polytechnic University, Afghanistan. The primary objective of these stations is to support the establishment of a reliable data basis of meteorological and hydrological data especially in remote areas with extreme climate conditions in Central Asia for applications in climate and water monitoring. Up to now ten years of data are provided for an area of scarce station distribution and with limited open access data which can be used for a wide range of scientific or engineering applications. The data described in this manuscript will be made publicly available with the DOI https://doi.org/10.5880/GFZ.1.2.2020.002 (Zech et al., 2020) after final acceptance. In the meantime, find data via this temporary link: https://kurzelinks.de/romps-data or via the Sensor Data Storage System (SDSS) at http://sdss.caiag.kg.

## 1. Introduction

Central Asia with its former Soviet republics of Kazakhstan, Kyrgyzstan, Tajikistan, Turkmenistan and Uzbekistan is a region that varies from high mountains to deep valleys, vast deserts and fertile river basins. Due to this wide range of natural diversity, large differences in climate forming factors like temperature, precipitation and snow cover occur.



Especially, the high mountains such as the Pamir and Tien Shan where most of the water originates from glaciers and snow packs provide important water resources for the entire region (Unger-Shayesteh et al., 2013). During the Soviet time, a large number of manually-controlled monitoring stations for meteorological and hydrological observations have been operated for extended periods of time. Specifically, the river discharge data was used to infer the melt-water of the snow covered mountains of Tajikistan and Kyrgyzstan with the demand for water in the arid but agricultural used land of Kazakhstan, Turkmenistan and Uzbekistan in exchange for coal, oil and gas (Bernauer and Siegfried, 2012). After the collapse of the Soviet Union in 1991, the sharing of resources became difficult and thus subject to frequent disputes between the now independent countries. Especially, the water availability and its different utilisation are the most challenging problems. While the water from the mountainous areas is used for the generation of hydropower due to the lack of other energy providing resources in the upstream countries, the demand for agricultural irrigation in the downstream countries in summer contrasts with the release of water from the reservoirs during the winter season (Siegfried et al., 2012). Additionally, the monitoring network degraded significantly after 1991 mainly due to economic shortening resulting in a lack of information urgently needed for water availability decisions (Unger-Shayesteh et al., 2015). To support the Central Asian countries in transboundary water resource management based on reliable in situ and remote sensing data and to make water a subject of peaceful cooperation, the German Federal Foreign Office launched the Central Asian Water Initiative ("Berlin Process") in April 2008. The primary goal was to assist the cooperation between the Central Asian countries with regard to energy and water management on the political, scientific-technical and educational level.

The scientific-technical level aimed at the establishment of a reliable data basis of hydrological and meteorological data and the implementation of new technical monitoring and data distribution solutions. These goals were addressed by the Regional Research Network „Water in Central Asia" (CAWa) funded by the German Federal Foreign Office and coordinated by the German Research Centre for Geosciences (GFZ) in Potsdam, Germany. To support the reconstruction of the degraded network of meteorological and hydrological station infrastructure and to provide near real-time hydrometeorological data, a network of Remotely Operated Multi-Parameter Stations (ROMPS) (Schöne et al., 2013) together with a remote-sensing monitoring system of rivers, reservoirs, and lake levels (Schöne et al., 2018a) has been established over the past ten years. To support the activities and to strengthen the geoscientific cooperation between the five Central Asian countries and Afghanistan, additional funding has been provided by GFZ through the "Global Change Observatory – Central Asia" (GCO) and the "Advanced Remote Sensing – Ground-truth Demo and Test Facilities" (ACROSS) projects of the Helmholtz Association (Helmholtz Society 2015). While some of these stations have been installed to additionally monitor the tectonically active parts of the Pamir and Tien Shan mountain areas with GNSS (Zubovich et al., 2016), others are dedicated to monitor glacier dynamics and Glacier Lake Outburst floods (Zech et al., 2016) and to re-establish glacier monitoring for mass balance studies (Hoelzle et al., 2012).





The objective of this paper is to describe the near-real time meteorological and hydrological
observations provided by the CAWa, GCO and ACROSS station network. An overview of the
station locations, the methods of data collection including their known quality issues and
further documentation is given in the next sections.

## 2. Station network

To support the construction of new hydrometeorological stations, close cooperation
between different agencies and institutes in Central Asia has been established. One of the
main partners is the Central Asian Institute for Applied Geosciences (CAIAG) in Bishkek,
Kyrgyzstan, which was founded in 2004 by the Government of the Kyrgyz Republic and the
GFZ. Additionally, cooperation with the national hydrometeorological services in Uzbekistan
(UzHM), Tajikistan (TjHM) and Kyrgyzstan (KgHM), the Ulug Beg Astronomical Institute
(UBAI) in Tashkent, Uzbekistan, as well as with the Kabul Polytechnic University (KPU),
Afghanistan, have been established. In joint collaboration with these partners, 18 stations
have been installed and are jointly operated altogether. Thereof eight stations from
Kyrgyzstan, three stations from Uzbekistan, two stations from Tajikistan and one station
from Afghanistan are providing data. Figure 1 shows the station distribution in Central Asia.

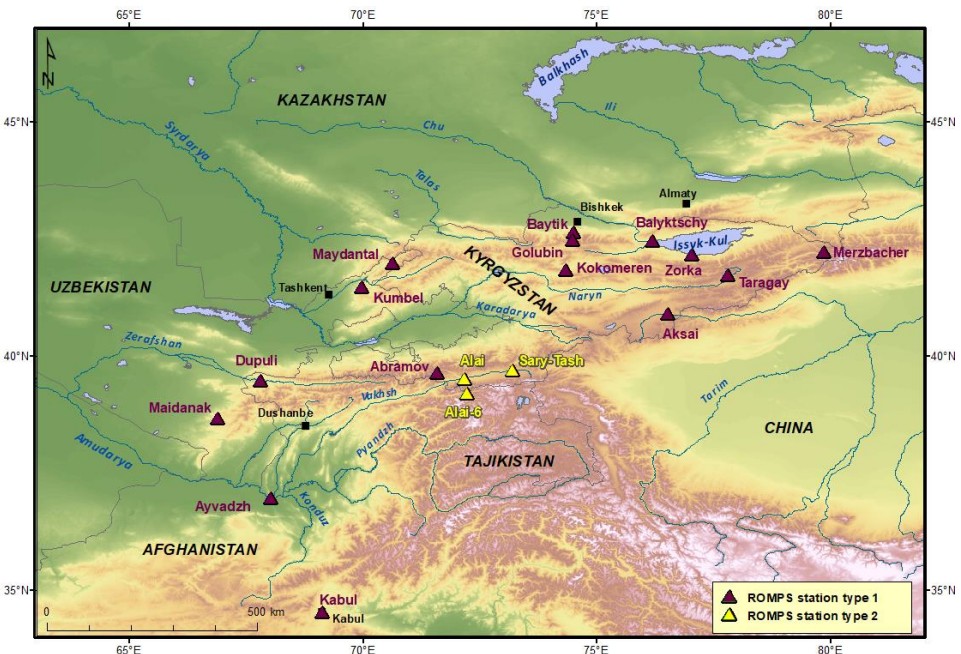

Figure 1: ROMPS network (credits to NOAA for the DEM)



The stations are located in different climatic regions, reaching from high altitudes (4124m a.s.l.) in the mountains and at glaciers down to low elevations (318m a.s.l.) in flat and dry areas covering three different Central Asian countries and Afghanistan (Schöne et al., 2018b; Schöne et al., 2019). The station locations were selected together with the Hydromet Services and partners with regard to previous network coverage, meteorological information content, possibility of satellite data transfer and station security aspects. However, some stations have been installed at existing meteorological stations or 'Hydroposts' with manual readings done by a local operator. The intention was to combine and compare the manual with the digital measurements as well as to ensure the safety of the equipment.

## 3. Instruments and Data Storage

The stations are designed for the operation in remote areas and high altitudes (see Figure 2 and Figure 3), especially under extreme climate conditions with temperatures ranging from +60°C to -60°C (Schöne et al., 2013). To keep the maintenance efforts low, the general technical setup of all ROMPS is identical at most locations. The system operates independently and automatically in order to reduce the need for manual interventions of a local operator. All stations consist of outside connected sensors and a water-proof (IP66) station main box (see Figure 6) integrating the central electronic components for the general operation such as the station computer system, independent (solar) power supply, and data communication systems. Two alternative communication lines were chosen, either two different satellite systems (VSAT and Iridium) or depending on the signal coverage one satellite (VSAT) and one GSM ground communication line. Due to their independence from local infrastructures (power, data transmission and manual interaction), the stations can be located in remote areas with difficult accessibility.

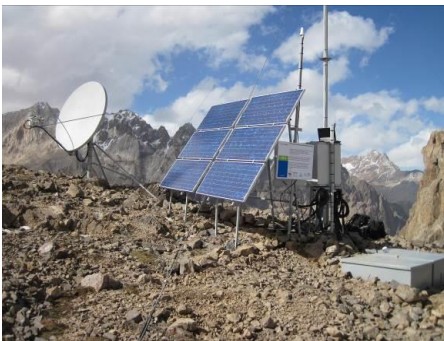
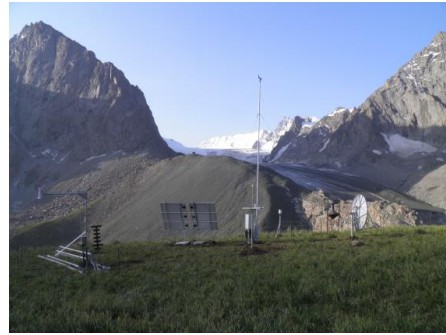

Figure 2: Station ABRA                    Figure 3: Station GOLU with snow system

The station outline is determined by the selection of meteorological sensors and can be differentiated into two types. The first type consists of a set of separate sensors complying with the WMO requirements (WMO, 2018) that are arranged around the station main box to

avoid interferences with each other. At the second type of station, a compact weather transmitter is used. Figure 4 shows a typical station setup for the first type of station but depending on the local environment, the sensors are arranged differently at other stations.

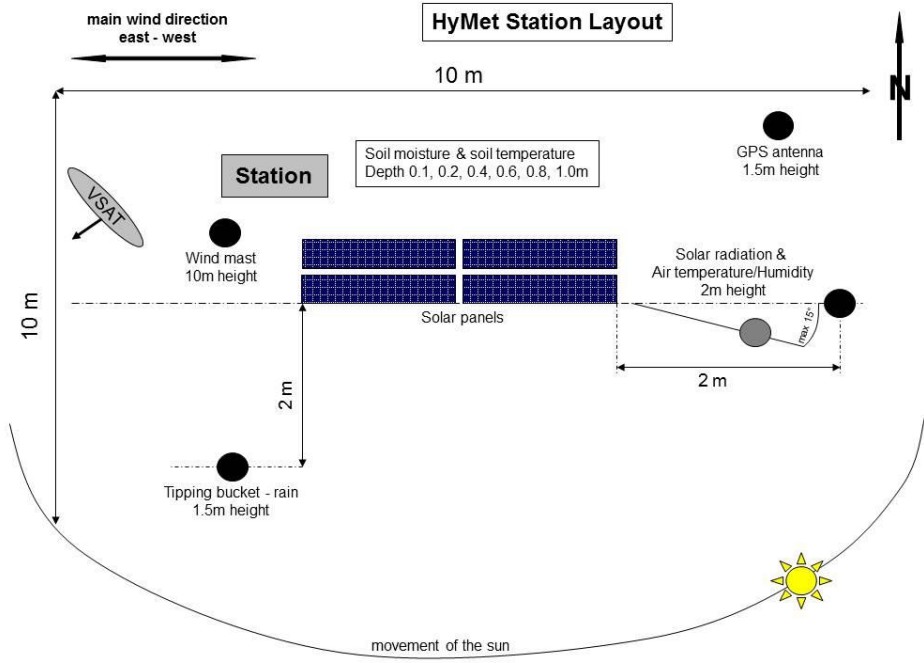

Figure 4: Typical station outline

All stations are equipped with a standard set of meteorological sensors. The following table lists the measured parameters for each station type:

Table 1: Measured parameters for each station type

| Type one | Type two |
|---|---|
| Air temperature and relative humidity | Air temperature and relative humidity |
| Barometric air pressure | Barometric air pressure |
| Wind speed and direction | Wind speed and direction |
| Precipitation | Precipitation and hail |
| Solar radiation | |
| Soil water content | |
| Soil temperature | |

At selected locations, the stations have been augmented with a river discharge monitoring system and/or a snow measuring system. While the snow system is usually located adjacent to the other sensors (see Figure 3 to the left), the discharge system is installed directly at rivers which can be several hundred meters away from the station main box (see Figure 5). It



consists of an independent power supply and the data is transmitted in regular intervals by a
radio link to the main station. The snow system measures the following parameters:

- snow depth (reversed distance),
- content of liquid water and ice,
- snow density, and
- snow water equivalent (SWE).

The river discharge system provides the parameters:

- water level,
- surface flow velocity, and
- (computed) river discharge volume.

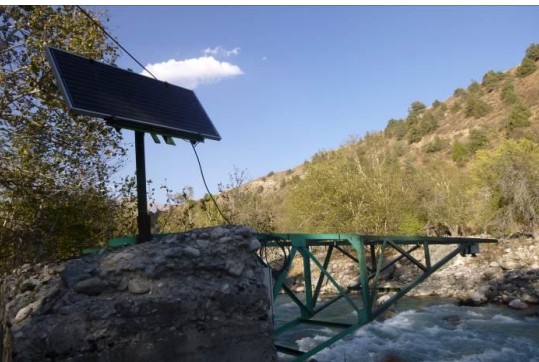

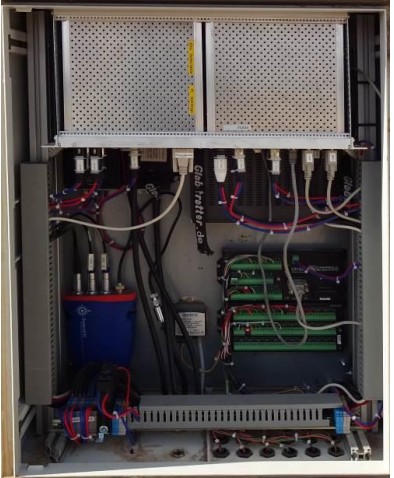

Figure 5: River discharge system at station MTAL with a solar panel and the boom for the sensor.

Figure 6: Station main box. Topside the computer and power module is visible. The VSAT and iridium modem are located in the back. Below is the GNSS receiver (left), the barometric sensor (middle) and the Campbell® datalogger (Schöne et al., 2013).

The primary service of the CAWa ROMPS network is to provide meteorological and hydrological data (Schöne et al., 2018b and 2019), especially from the high mountain areas of Central Asia supporting the national Hydrometeorological services and the regional and international scientific community. Additional to the hydrometeorological data acquisition, some stations integrate other sensors, such as broadband seismometers for the GEOFON network (GEOFON, 2020), automated cameras for glacier monitoring and mass balance calculations (Hoelzle et al., 2017), GNSS receivers for investigation of glacier dynamics and Glacier Lake Outburst Flood (GLOF) monitoring (Zech et al., 2016 and 2018) or





measurements of snow water equivalent (SWE) with Cosmic-Ray Neutron Sensing (e.g., Schattan et al., 2018). The stations with their installed sensor systems and start of operation are listed in Table 2. The start of operation does not necessarily match with the start of all sensor systems. In some cases, individual sensors were installed at a later date.

Table 2: Station list with beginning of operation dates and installed sensors systems. While the column
'meteorological sensors' refer to various individual sensors, the 'weather sensor' column refers to the compact weather transmitter (station type two).

| Station | Country | Partner | Name | Start of operation | Meteo. Sensors (type 1) | Weather Sensor (type 2) | River discharge | Snow para-meters |
|---------|---------|---------|------|--------------------|-------------------------|-------------------------|-----------------|------------------|
| ABRA | KGZ | CAIAG | Abramov Glacier | 08/2011 | + | - | - | - |
| ALAI | KGZ | CAIAG | Alai | 05/2017 | - | + | - | - |
| ALA6 | KGZ | CAIAG | Alai-6 | 10/2015 | - | + | - | - |
| ASAI | KGZ | CAIAG | Aksai | 07/2012 | + | - | - | - |
| AYVA | TAJ | TjHM | Ayvadzh | 06/2012 | + | - | - | - |
| BALY | KGZ | CAIAG | Balyktschy | 06/2017 | + | - | - | - |
| DUPU | TAJ | TjHM | Dupuli | 05/2012 decom. 12/2012 | + | - | + | - |
| GOLU | KGZ | CAIAG | Golubin Glacier | 09/2013 | + | - | - | + |
| HM01 | KGZ | CAIAG | Baytik | 12/2010 | + | - | - | + |
| KABU | AFG | KPU | Kabul | 04/2015 | + | - | - | - |
| KEKI | KGZ | CAIAG | Kokomeren | 11/2010 | + | - | + | - |
| KMBL | UZB | UzHM | Kumbel | 09/2015 | + | - | - | + |
| MADK | UZB | UBAI | Maidanak | 11/2012 | + | - | - | - |
| MRZ1 | KGZ | CAIAG | Merzbacher | 08/2011 | + | - | - | - |
| MTAL | UZB | UzHM | Maydantal | 09/2014 | + | - | + | + |
| OYGA[1] | UZB | UzHM | Oygaing | 10/2018 | - | - | + | - |
| SARY | KGZ | CAIAG | Sary-Tash | 08/2015 | - | + | - | - |
| TARA | KGZ | CAIAG | Taragay | 09/2010 | + | - | - | - |
| ZOKA | KGZ | CAIAG | Zorka | 09/2016 | + | - | - | - |

### 3.1 Meteorological sensors

The compact Weather Transmitter (Vaisala WXT520 or WXT530) offers six meteorological
parameters in one compact unit. It measures wind speed and direction, precipitation, atmospheric pressure, air temperature and relative humidity. The wind is estimated by transit time through three equally spaced ultrasonic transducers on a horizontal plane. The precipitation sensor on top of the transmitter, detects the impact of individual raindrops by noise detection and thus, the accumulated rainfall, the rain intensity and duration. It is
capable to distinguish between rain and hail. The PTU (pressure, temperature, humidity)

---

[1] OYGA (Oygaing) consists of a river discharge system only that is located close to and connected to the station MTAL (Maydantal).



module measures the atmospheric pressure, air temperature and humidity with a capacitive silicon element for the pressure, a capacitive ceramic element for the temperature and a capacitive thin film polymer element for the humidity. The PTU component is fixed insight a radiation shield to protect the sensor from direct sun light.

The combined Temperature and Humidity Sensor (Vaisala HMP45, Vaisala HMP155, Campbell® Scientific CS215 or Galltec+Mela KPK 1/5-ME) provides air temperature and relative humidity data. The humidity measurement is based on the capacitive thin-film polymer probe which either absorbs or releases water vapour that changes the dielectric properties and, therefore, the capacitance of the sensor. The temperature measurement is

based on the resistive platinum probe. Both probes are located at the tip of the sensor and protected by a sintered Teflon filter. The sensor is fixed inside a radiation shield to protect the sensor from direct sun light and is mounted on a steel mast.

The Barometric Pressure Transducer (Setra 278 or Campbell® Scientific CS115) measures the actual local atmospheric pressure. The sensor is fixed inside the station main box but is

conducted outside of the box with a connecting air tube.

The alpine Wind Monitor (RM Young 05103-45) is used to measure the horizontal wind speed and direction. The wind speed is measured with a helicoid-shaped, four-blade propeller. The rotation of the propeller produces a signal proportional to the wind speed. The position of the vane is transmitted by a potentiometer and its output voltage is

proportional to the wind direction. The wind sensor is mounted on a steel mast at 10m above ground.

The Tipping Bucket Rain Gauge (RM Young 52203 or Thies 5.4032.35.008) measures liquid precipitation with a tipping bucket mechanism. The measured liquid rain drains through a collection tube for verification of the total rainfall by mechanically tipping a scale. As the

tipping bucket is not heated, winter precipitation is strongly biased in regions with frequent temperatures around or below 0°C. The rain sensor is installed on a steel mast which is fixed with tensioning wires to reduce the influence of wind induced vibrations.

The Net Radiometer (Hukseflux NR01) measures the energy balance between the incoming short-wave and long-wave infrared radiation versus the surface-reflected short-wave and

outgoing long-wave infrared radiation. It consists of a pyranometer and a pyrgeometer pair that faces upward and a complementary pair that faces downward. The pyranometers and pyrgeometers measure the short-wave and the far infrared radiation, respectively. The sensor is fixed at a steel mast.

The Water Content Reflectometer (Campbell® Scientific CS616) measures the volumetric

water content of porous media. It uses time-domain measurement methods that are sensitive to the dielectric permittivity of the used medium which changes with the water content of the soil material. In our installations, typically six sensors are installed in the ground at different depths.



The Soil Temperature Probe (Campbell® Scientific T107) uses a thermistor to measure the temperature and, therefore, can be used in air, water, and soil. Typically, six sensors are installed in the ground in different depths adjacent to the Water Content Reflectometers.

### 3.2   Hydrological sensors

The Discharge Measurement System (Sommer RQ24) enables the contact-free measurement of the surface flow velocity plus the water level and automatically calculates the discharge quantity of the water using a user-defined river cross section. The measurement of flow velocity is based on the Doppler Shift principle. The sensor calculates local flow by comparing an emitted frequency with the frequency reflected by the water surface (Doppler shift). The water level is measured by using time-delay measurements. The pulse radar emits impulses with a specific length in the lower micro-second range perpendicular to the water surface. The time delay between emission and reception of the impulse is proportional to the distance from the water surface. The calculation of the discharge is based on the continuity equation and determined from the cross section depending on the measured water level, the measured surface velocity and a pre-defined related k-factor which represents the hydraulic properties of the river (e.g., roughness of river bed). While the water level and surface velocity are measured continuously, the cross-section and k-factor are pre-defined during the installation or changed during maintenance visits.

The Snow Pack Analyzer (Sommer SPA) in combination with a Snow Depth Sensor (Sommer USH8) provides the different parameters of snow like snow density, snow water equivalent (SWE) as well as contents of liquid water and ice. The snow depth is measured with an ultrasonic pulse as a distance between the sensor and the (snow) surface. To estimate the volume contents of the individual snow elements, the complex impedance along a flat ribbon sensor is measured as different components in the snow pack (ice, water, air) which have different dielectric constants. The specific volume contents and the liquid water and ice content in the snow are used to calculate the snow density. A combination of the snow depth and the snow density defines the SWE. The content of liquid water and ice is measured at different positions in the snow pack with one sloping sensor (sensor 1) and three horizontal sensors with typical positions of 10 cm (sensor 2), 30 cm (sensor 3), and 50 cm (sensor 4) above ground. Therefore, the sensor provides four different values for the snow density, the SWE, and content of liquid water and ice. The USH8 snow depth sensor can also be operated independently without the SPA. Then, the snow depth is measured only.

Due to the remoteness of most stations, regular calibrations of sensors could not be performed. But sensors have been exchanged when problems occurred. Notes to the station documentation can be found in chapter 6.




## 4. Datasets

The datasets comprise the data from all stations and their different sensor systems. The meteorological data has been sampled in 1-minute intervals and then converted (average "_Avg", maximum "_Max", time of maximum "_TMax", minimum "_Min", or total "_Tot") to 5-minute data that is stored in meteorological files separated for each station. The time consistency is achieved by daily synchronizing the system with a GPS time signal.

### 4.1 Meteorological data

Table 3 lists all meteorological parameters for the first type of station (individual sensors) for each parameter with their abbreviation, unit and type of sampling to a 5-minute value.

Table 3: List of meteorological parameters measured by the individual sensors (station type one)

| Measurement parameter | Description | Unit | Type of sampling |
|---|---|---|---|
| AirTC | Air temperature | °C | Sample |
| RH | Relative humidity | % | Sample |
| Baro | Barometric air pressure | hPa | Sample |
| WindSp_Avg | Wind speed | m/s | Average |
| WindSp_Max | Wind speed maximum (Gust) | m/s | Maximum |
| WindSP_TMax | Time of wind speed maximum | Date and Time | Date and Time of Maximum |
| WindDir | Wind direction | ° (degree) | Sample |
| Rain_Tot | Precipitation | mm | Total |
| RadSW_Up_Avg | Incoming short-wave solar radiation | W/m$^2$ | Average |
| RadSW_Dn_Avg | Outgoing (reflected) short-wave solar radiation | W/m$^2$ | Average |
| RadLW_Un_Avg | Incoming long-wave solar radiation | W/m$^2$ | Average |
| RadLW_Dn_Avg | Outgoing (reflected) long-wave solar radiation | W/m$^2$ | Average |
| NR01TC_Avg | Temperature at the solar radiation sensor in degrees Celsius | °C | Average |
| NR01TK_Avg | Temperature at the solar radiation sensor in Kelvin | K | Average |
| NetRs_Avg | Net short-wave solar radiation | W/m$^2$ | Calculated[2] |
| NetRl_Avg | Net long-wave solar radiation | W/m$^2$ | Calculated[2] |
| Albedo_Avg | Proportion of the incident light or radiation that is reflected by a surface | | Calculated[2] |
| UpTot_Avg | Total incoming solar radiation | W/m$^2$ | Calculated[2] |
| DnTot_Avg | Total outgoing solar radiation | W/m$^2$ | Calculated[2] |
| NetTot_Avg | Total Net solar radiation | W/m^$^2$ | Calculated[2] |
| RadLW_UpCo_Avg | Temperature corrected incoming long-wave solar radiation | W/m$^2$ | Calculated[2] |
| RadLW_DnCo_Avg | Temperature corrected outgoing (reflected) long-wave solar radiation | W/m$^2$ | Calculated[2] |

[2] These values are calculated by the datalogger acoording to the manufacturer's instructions.





| VW_#[3] | Volumetric soil water content at local position | n/a | Sample |
|---|---|---|---|
| PA_#[3] | Measured travel time of the EM-wave along the probe at local position | µSec | Sample |
| T107_#[3] | Soil temperature at local position | °C | Sample |

Within the datasets of all meteorological sensors, additional technical values (e.g., battery voltage, record number) are provided but not listed in Table 3 and Table 4. Detailed information can be found in the data format specification which is part of the supplementary
material. Table 4 lists all meteorological parameters for the compact weather transmitter (station type two).

Table 4: List of meteorological parameters measured by the compact weather transmitter (station type two)

| Measurement parameter | Description | Unit | Type of sampling |
|---|---|---|---|
| Ta | Air temperature | °C | Sample |
| Ua | Relative humidity | % | Sample |
| Pa | Barometric air pressure | hPa | Sample |
| Dn | Wind direction minimum | ° (degree) | Minimum |
| Dm | Wind direction average | ° (degree) | Average |
| Dx | Wind direction maximum | ° (degree) | Maximum |
| Sn | Wind speed minimum | m/s | Minimum |
| Sm | Wind speed average | m/s | Average |
| Sx | Wind speed maximum | m/s | Maximum |
| Rc | Rain accumulation | mm | Total |
| Rd | Rain duration | s | Time |
| Ri | Rain intensity | mm/h | Total |
| Hc | Hail accumulation | hits/cm$^2$ | Total |
| Hd | Hail duration | s | Time |
| Hi | Hail intensity | hits/cm$^2$h | Total |

**4.2 Hydrological data**

The snow parameters are measured every 15 minutes and stored in files, separately from the meteorological data. Table 5 lists the parameters of the snow system that are mainly used for hydrological studies. The system provides additional technical parameters to control the system. A detailed description can be found in the data format specification
which is part of the supplementary material.

Table 5: List of snow parameters

| Name of parameter | Description | Unit |
|---|---|---|
| SH | Snow depth | m |

---

[3] With #: Sensor 1..6

| S#_dens[4] | Snow density at position # | Kg/m$^3$ |
| S#_SWE[4] | Snow water equivalent (SWE) at position # | mm |
| S#_ice[4] | Content of ice at position # | % |
| S#_water[4] | Content of liquid water at position # | % |

The river discharge system performs several consecutive scans of the water surface and checks the reflected value for spikes and weak signals before providing the measurement result. This can cause different measurement intervals depending on the turbulence of the water surface. Like the snow parameters, the discharge measurements are stored in separated files. Table 6 lists the hydrological parameters of the river discharge system. The system provides additional technical parameters to monitor the system state. The detailed description can be found in the data format specification which is part of the supplementary material.

Table 6: List of river discharge parameters

| Name of parameter | Description | Unit |
|---|---|---|
| R_WL | Water level | mm |
| R_vel | Surface flow velocity | mm/s |
| R_Q | Calculated river discharge | m$^3$/s |

### 4.3 Sensor Data Storage System

The easy and open-access provision of meteorological and hydrological data has been the main objective of the ROMPS network operation. The data is open for the usage in environmental research, public information services such as the Hydromet services provide, and to support information based decision-making processes especially in the fields of water and land management and climate adaption. The files coming from the stations are forwarded directly to the open-access Sensor Data Storage System (SDSS) developed and hosted at CAIAG in Bishkek, Kyrgyzstan, and are therefore immediately available to the public. A graphical user interface (in English, Russian, and German language) offers the possibility to request the data interactively by selecting particular stations on a map or from a list of parameters. These values can be displayed as time series and downloaded as an Extensible Markup Language (XML) file. The SDSS web-page is accessible through the following link: http://sdss.caiag.kg.

### 4.4 Data completeness

All stations are supplying data continuously since they were installed. But it should be noted that most of the stations are located in remote areas and could not be visited regularly or immediately after technical problems occurred. Therefore, failover procedures to check for

---

[4] With #: Sensor 1..4



stalled software programs and automatic restart scripts have been implemented to minimize or avoid resolvable errors or gaps. Nonetheless, hardware cannot always be secured against outages, especially in remote areas. Major data problems or severe sensor failures that emerged are listed in section 5.

Additionally, due to errors in the datalogger configuration, the data was not always sampled 295 in five minute intervals. At some stations, the data was sampled every minute or with other intervals which especially has an impact to the rain value and the wind speed maximum (gust) that should be a total over five minutes instead of, e.g., one minute.

## 5. Data Quality

The available data has to be considered as raw data coming directly from the stations and 300 have not undergone any quality control (QC). The primary purpose of the network is to provide near real-time data for the Hydromet services without major time delay. A consistency or QC on this dataset is beyond the scope of the network operation. QC is supposed to be done at each Hydromet service individually as they are the responsible national agency for international data exchange or accordingly, for each single user.

### 5.1 General problems

Nevertheless, there are known quality issues that emerged during the operation of the station and are mentioned in this paper to support further user quality management procedures. Especially the known problems such as incorrect configurations, wrong sensor installations or sensor failures that are not obvious for data users are explained below.

The rain sensor with a tipping bucket measurement principle is installed on a steel pillar which tends to vibrate during strong gust. This can cause the tipping beam to tilt and hence falsely produce a tip, particularly in areas exposed to strong winds. As a consequence, all rain sensors have been equipped with additional supporting crossbars to avoid this problem during the years 2013 and 2016. In the winter time, the measurement of snow or the 315 differentiation between snow and rain is challenging as the sensor is not heated. While parts of the snow in the sensor's cone evaporate, other parts melt and are registered as a measurement. Furthermore, the water from the rain drops has to pass a small funnel to reach the tipping bucket. This funnel can become blocked due to leaves or other dirt which may hamper the measurements and can lead to underestimated or delayed rain data. As 320 there are no local operators at most stations or they are not living close to the station, the rain sensors could be cleaned only during maintenance visits.

The soil sensors tend to have jumps in the measurements with so far unknown reasons and different time spans although the sensors haven't been touched. In some areas, these sensors are affected by animals (e.g. at TARA by marmots) that nibble the cables. These 325 jumps and sensor failures will not be mentioned in the following listings as these erroneous data is clearly identifiable.

All calculated values (see Table 3) of the solar radiation sensor but specifically the albedo provides suspicious values close to sunrise and sunset. The albedo is calculated by the datalogger as the quotient from the reflected and the incoming short-wave radiation. This

can lead to erroneous results when the incoming values tend to be close to zero when it is (nearly) dark.

### 5.2 Problems in the meteorological measurements

Most problems resulted from sensor failures, errors or inconsistencies in the incorrect configuration, or incorrect technical installations of sensors. Table 7 lists all known problems

of meteorological sensors at the stations. Erroneous data has not been removed from this data set as it is the raw data coming directly from the station without any quality control. Details about sensor changes or cleaning of sensors can be found in the station documentation (section 6).

Table 7: List of problems in the meteorological data

| Station | System | Parameter | Description of error | Time |
|---|---|---|---|---|
| ASAI | Rain | Rain_Tot | Sensor failure, rain could not be measured | Until 11.07.2017 |
| | Solar radiation | RadLW_Dn | Sensor failure, values incorrect | 11.07.2017-04.10.2018 |
| BALY | Tempe-rature | AirTC | Incorrect configuration | Until 17.05.2018 |
| HM01 | Tempe-rature | AirTC | Sensor failure, values incorrect | 23.06.2012-02.10.2013 |
| | Humidity | RH | Sensor failure, values incorrect | 23.06.2012-02.10.2013 |
| | Air pressure | Baro | Incorrect configuration | Until 11.07.2012 |
| | Wind | Wind_Dir | Incorrect configuration | Until 09.07.2010 |
| KEKI | Air Pressure | Baro | Incorrect configuration | Until 18.04.2013 and 05.07.2017 -14.03.2019 |
| KMBL | Rain | Rain_Tot | Wrong sensor installation, no values | Until 07.10.2018 |
| MRZ1 | Tempe-rature | AirTC | Sensor failure, values incorrect | 01.05.2014-08.07.2015 |
| | Humidity | RH | Sensor failure, values incorrect | 01.05.2014-08.07.2015 |
| | Wind | Wind_Sp, Wind_Dir | Sensor broken, no values | 19.05.2015-08.07.2016 |

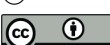


| | Rain | Rain_Tot | Sensor broken, no values | Unclear[5] - 08.07.2016 |
| | Rain | Rain_Tot | Sensor broken, no values | Unclear[6] - 23.08.2019 |
| TARA | Tempe-rature | AirTC | Sensor failure, values incorrect | 14.07.2012- 26.06.2013 |
| | Humidity | RH | Sensor failure, values incorrect | 14.07.2012- 26.06.2013 |
| | Air pressure | Baro | Incorrect configuration | Until 13.07.2012 |
| | Solar radiation | RadSW_Up | Sensor failure, values incorrect | 31.08.2016- 15.06.2017 |
| | Solar radiation | RadLW_Up | Incorrect configuration | Until 11.09.2012 |
| | Solar radiation | RadLW_Dn | Sensor failure, values incorrect | 09.22.2017- 15.06.2017 |


## 5.3 Problems in the hydrological measurements

Beside incorrectly configured or connected sensors, problems with the snow depth sensor (values SH) occurred when referencing the time-delay distance measurement to a zero level of the snow or ground, respectively. Due to the accuracy of the system of ±4mm and the condition of the ground reflection during the referencing process (e.g. fresh vegetation in spring or flat and dry in autumn), negative values close to zero can occur. Furthermore during the summer season at some places, growing grass can affect the measurement by producing a feigned snow depth. The signals of growing grass and sometimes the cutting of grass is clearly visibly as well as the time delay between melting of the snow and growing of grass or first snow in autumn, respectively.

The calculated discharge values provided by the river discharge system are the combination from the measurements of the water level and surface velocity with the cross profile of the river and a k-factor which accounts for the hydraulic model of the river flow. Especially, the river cross profile could not always be determined directly at the sensor's installation side, but only some meters away. In addition, the cross section profile may change over time leading to different k-factors. Therefore, these values should be used with caution. The discharge values are always influenced by the inaccuracy of the cross profile of the river. Table 8 lists the known problems with the snow and the river discharge system.

Table 8: List of problems in the hydrological parameters

| Station | System | Parameter | Description of error | Time |
|---------|--------|-----------|---------------------|------|
| GOLU | Snow | All | Sensors connected incorrectly | Until 11.06.2014 |

---

[5] May have been broken at any time between the maintenance visits on 24.07.2014 and 08.07.2015.
[6] May have been broken at any time between the maintenance visits on 29.07.2017 and 13.08.2019.



| KMBL | Snow | SH | Sensor connected incorrectly | Until 05.05.2016 |
|---|---|---|---|---|
| KMBL | Snow | S1_ice, S1_water, S1_dens, S1_SWE | Sensor failures during winter time | Winter 2017, 2019 |
| KEKI | River | All | Sensor failure | 12.7.17-20.9.17 |
| MTAL | River | R_vel, R_Q | During low water levels, a big stone appeared in the sensor's measuring footprint. The water flow became turbulent causing erroneous measurements. The problem was fixed on 26.09.2019. | Autumn until spring in 2016, 2017 and 2018 |
| OYGA | River | R_vel, R_Q | Test measurements during installation, not representative for the river | 14.10.2018 |


These are only the known and documented errors. This list is not intended to be exhaustive. For further usage of the data, different quality control procedures following WMO or national standards should be implemented. An outlook and suggestion gives section 8. Additionally, more detailed information about the stations and installed sensors is provided
by the station documentation (section 6).

## 6. Station documentation

For each station an extensive documentation exists describing the location and its surrounding, the technical installation of the station and the local conditions which might be useful to further interpret certain variations in the sensor data. As the installations had to be
adjusted to the local environment, installation maps and specific sensor installation (e.g. height/depth of sensors) are included in the documentation. Due to the spatial expansion and the difficult accessibility, maintenance activities could not be performed on a regular basis. Therefore, the documentation also lists all station visits and exchanges of sensors. The file containing the documentation is named as follows XXXX-HMT-SED-GFZ.pdf with

• XXXX: 4-letter code for station name,
       • HMT: HyMet station,
       • SED: Station Exposure description, and
       • GFZ: Agency of originator

These files are part of the supplementary material.

**7. File Name Convention and Data Format**



### 7.1 File names

All data are stored in files in ASCII format containing typically one hour of data (with sometimes different storage intervals). The different data types are separated into different files. The file name convention provides unique identifiers to distinguish between the
different types of measurements as follows:

`XXXX-<type-of-data>-<timestamp>.log` with

- XXXX:          4-letter leading identifier for station abbreviation (see Table 2),
- \<type of data>: hymetd       for meteorological parameters (station type one),
                  meteod       for meteorological parameters (station type two),
390                  RQ24         for river discharge parameters,
                  snow         for snow parameters,
                  USH8         for snow depth values, and
- \<timestamp>:  Unix time (seconds since 01.01.1970).

Due to historical reasons, a second file name convention exists which is used for the second type of stations. These files are named as follows:

`XXXXWWWWD.met` with

- XXXX:          4-letter leading identifier for station abbreviation (see Table 2),
- WWWW:       GPS week (weeks since 06.01.1980), and
- D:             Day of week with Sunday=0.

The individual data files are packed into an archive with tar (Petersen, 2007) that contains one month of data files. This monthly file is additionally compressed with bzip2 (Petersen, 2007) to reduce the memory requirements.

The monthly archive files are named as follows:

`XXXX-<type-of-data>-<year>-<month>.tar.bz2` with

- XXXX:          4-letter leading identifier for station abbreviation (see Table 2),
- \<type of data>: hymetd       for meteorological parameters (station type one),
                  meteod       for meteorological parameters (station type two),
410                  RQ24         for river discharge parameters,
                  snow         for snow parameters,
                  USH8         for snow depth values,
- \<year>:       4-digit year, and
- \<month>:     2-digit month of the year.

After one year, these twelve monthly files are further archived and compressed to one file. The yearly files follow the same file name convention as the monthly files but the month is missing in the file name.



```
XXXX-<type-of-data>-<year>.tar.bz2.
```

**7.2   Data format**

Two data formats exist for the different station types. As the sensors for station type one are connected to a Campbell® data logger, the file format largely corresponds to the ASCII Campbell® data format. It can be separated into header and data sections. The header section can be further divided into four parts containing one line each as follows:

1.   General station information,
      2.   Types of measured parameters listed with their abbreviation,
      3.   Units of measured parameters, and
      4.   Quantity representation (type of sampling) of measured parameters.

After this header section, the data section follows starting with date and time. All values
corresponding to the same measurement time are written in one line and are comma-separated. A detailed description of all parts of the data format can be found in the data format specification (CAWA-SSP-FMT-GFZ-006.pdf) which is part of the supplementary material.

The sensor at station type two provides data from a compact VAISALA weather transmitter
retrieved directly to the PC by dedicated software. Therefore, the ASCII data format is different but can be also divided into a header and a data section.

The header section provides information about the following parameters:

      1.   Requesting program name and version,
      2.   Date and starting time of the measurements in this file,
3.   Sensor type, and
      4.   Sampling rate.

After this section, the data section follows with the time of the measurement. All values are comma-separated but are divided into several lines depending on the meteorological parameter.  A detailed description of all parts of the data format can be found in the data
format specification (GITW-SSP-FMT-GFZ-003.pdf) which is part of the supplementary material.

## 8.   Outlook

To monitor the quality of the sensor data prior to their use in computation of climate variables, basic QC procedures should be applied. In accordance with international
guidelines on QC procedures (WMO, 2017; WMO, 2004; WMO, 2018) different levels of QC procedures are suggested and should be considered before using the data. The provided data can be seen as raw data without any QC implied. We recommend performing the following QC steps before using the data:



1. Integrity and syntax check:

Test to search for gross errors in the data (e.g. transmission or data storage errors like wrong characters).

2. Plausibility check (tolerance test for each sensor):

All instantaneous values shall be checked against configurable range limits (e.g. sensor range limit specification). These limits are different for each

meteorological/hydrological parameter.

3. Time consistency check:

Check of the rate of change to the previous value, to test for the maximum allowed variability.

4. Persistence check:

Check of the rate of change to the previous values, to test for the minimum required variability.

5. Internal consistency check

Test to check parameters against each other for plausibility.

Initial efforts have already been made to implement the first two tests in the SDSS but the

remaining tests have not been developed, yet. Additionally, the authors are preparing a publication of a quality controlled data set at a later time.

## 9. Data availability

The data described in this manuscript will be made publicly available as monthly data files separated for each station with the DOI https://doi.org/10.5880/GFZ.1.2.2020.002 (Zech et al., 2020) after final acceptance. In the meantime, find data via this temporary link: https://kurzelinks.de/romps-data. Additionally, the near real-time data can be displayed and downloaded without any registration from the user interface SDSS at http://sdss.caiag.kg.

## 10. Summary

In Central Asia, the access to hydrometeorological data especially from remote areas is still

limited. Within the CAWa project funded by the German Federal Foreign Office and with the support of the "Global Change Observatory – Central Asia" (GCO-CA, GFZ) and the ACROSS (Helmholtz Association) initiative, a network of remotely operated multi-parameter stations (ROMPS) have been installed in Kyrgyzstan, Uzbekistan, Tajikistan, and Afghanistan. The technical concept has proven to withstand harsh and varying climate conditions without the

necessity of permanent human interaction. The data presented in this paper is the result of the operation of these stations in the past ten years. The stations provide raw hydrometeorological information such as air temperature, relative humidity, air pressure, wind speed and direction, precipitation, solar radiation, soil moisture and soil temperature as well as snow and river discharge information where available. The data can be used for

different applications ranging from scientific investigations of climate change, ground-



truthing of remote sensing based technologies and improvement of weather forecasts, to more politically based decisions on water management considerations and climate adaption strategies.

**Supplement**

The supplementary material to this article consists of the data format specification and the station documentation for each station. As most stations are operational, changes in these documents will occur but will be specified in the change log of each document. The supplementary material will be provided with the DOI https://doi.org/10.5880/GFZ.1.2.2020.002 (Zech et al., 2020) after final acceptance. In the meantime, it can be accessed via this temporary link: https://kurzelinks.de/romps-data.

**Author contributions**

CZ organized several fieldworks for station installation and maintenance works, compiled the data archives, the known quality issues and the supplementary documentation with support by JI and TS. The continuous station operation was done jointly by CZ, JI and TS. NS, TQ, MK and HT prepared and assembled the technical parts of the system and supported the on-site

technical installations. AZ designed and implemented the SDSS and keeps it operational. The partners from the Central Asian Countries Kyrgyzstan, Uzbekistan, Tajikistan and Afghanistan provided the necessary permits for the station installations, essential support for building and maintaining the stations and continuous data transmission during the last years in their countries. This applies in particular to AZ and AS for the stations ABRA, ALAI, ALA6, ASAI,

BALY, GOLU, HM01, KEKI, MRZ1, SARY, TAR and ZOKA in Kyrgyzstan, KZ and KT for the stations MTAL, OYGA and KMBL in Uzbekistan, YT for the station MADK in Uzbekistan, SO for the stations DUPU and AYVA in Tajikistan, and ZP for the station KABU in Afghanistan. KUS drafted a very first version of this paper many years ago. AG fostered the cooperation and negotiations especially between UzHydromet and GFZ and supported the fieldworks. BM

enabled the numerous station installations in Kyrgyzstan by providing support from CAIAG. CZ wrote this paper with essential support by TS. All authors revised and approved the content of the paper.

**Competing interests**

The authors declare that they have no conflict of interest.

**Acknowledgements**

The presented data and activities have been generously funded in the frame of the CAWa project (http://www.cawa-project.net) by the German Federal Foreign Office as part of the "German Water Initiative for Central Asia" (the so-called "Berlin Process", grant AA7090002). Additional funding for the installation and the long-term operation of the monitoring

network was and is being provided by the GFZ German Research Centre for Geosciences in the frame of the "Global Change Observatory - Central Asia" and the "Advanced Remote



Sensing – Ground-truth Demo and Test Facilities" (ACROSS) projects of the Helmholtz Association.

We appreciate the close cooperation with partners such as the Central Asian Institute for Applied Geosciences (CAIAG) in Kyrgyzstan, the National Hydrometeorological Services of Kyrgyzstan, Tajikistan and Uzbekistan, the Ulugh Beg Astronomical Institute at the Academy of Sciences of the Republic of Uzbekistan and the Kabul Polytechnic University of Afghanistan. Their generous support and help made the site selections and station installations possible. In particular, at CAIAG the support of Abdysamat Shakirov, Mikhail Borisov, Talant Altynbek uulu, and Zholdoshbek Okoev is highly appreciated. We also thank Alexander Merkushkin for various discussions and the help during the Abramov installation and his colleagues Tokhir Gafurov, Evgeniy Pavlov, Vasiliy Proxorov, Dmitriy Soloyd, Djura Sadikov, Anatholy Skorokhodov and Olga Mokh of UzHM (Uzbekistan) for supporting all station activities in Uzbekistan. From TjHM we appreciate the help and support from Muhiddin Yakubov.

In addition, we would like to thank the researchers of the University of Fribourg (World Glacier Monitoring Service) for the inspiring cooperation and interesting discussions, in particular Martin Hoelzle for the proposal to integrate an optical camera system for glacier monitoring at Abramov glacier and his team including Martina Barandun and David Sciboz for the support during and after the installation of the Abramov and Golubin stations. This work is supported through the Capacity Building and Twinning for Climate Observing Systems Program (CATCOS) of the Swiss Agency for Development and Cooperation. Furthermore, the help and support of Najibullah Kakar of the Norwegian Afghanistan Committee for the already installed as well as the station which is in progress is highly appreciated. At GFZ, we would like to thank the GEOFON team and all other colleagues who supported the station preparation and various travels to Central Asia.

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
