# Peer review of "Hydrometeorological Data from a Remotely Operated Multi-Parameter Station network in Central Asia"

_Earth System Science Data, 2020_

## Referee Comment (RC1) · Anonymous Referee #1 · 21 Oct 2020

The manuscript is well written and describes in solid technical detail the benefits and challenges of a remotely operated hydrometeorological observation network in a high altitude region with generally difficult climatic and accessibility conditions. The details of installed the systems and sensors, as well as some of the challenges faced to maintain the observations, are useful lessons. In addition, the access to the resultant data will be of great use to a range of climate researchers.

While recognizing the primary objective of the observing network is for international research purposes, the question arises with regards to its institutional ownership with the national operational agencies of the hosting countries. Are these observations in-

tegrated into the existing observation networks, for example of the hydrometeorological agencies? Considering the international funding of the observing network, should the observations not be made publicly available in real- or near-real-time (and if appropriate submitted to the WMO GTS to provide global, regional and national benefits)? If the international funding terminates, will the countries be able to continue to operate and maintain the systems?

These are important open questions that arise when reading the manuscript, in terms of sustainability and long-term impact. However the lack of information on these issues does not significantly undermine the quality of the manuscript.

---

## Author Comment (AC1) · 3 Nov 2020

We thank reviewer #1 for the constructive comments and suggestions. Concerning the three comments, the authors answer as follows: Comment 1: Are these observations integrated into the existing observation networks? The stations in Central Asia are installed in cooperation with the national Hydrometeorological Services. The data is made immediately accessible to them and is used for their operational purposes.

Comment 2: Should the observations not be made publicly available in real- or near-real-time (and if appropriate submitted to the WMO GTS to provide global, regional and national benefits)? The stations are property of the national agencies and, there-

fore, the Hydrometeorological Services are in charge of integrating the data into international data streams like the WMO. The data is also made available without delays through the SDSS system and offers machine-based Web-requests.

Comment 3: Will the countries be able to continue to operate and maintain the system? The long-term operation of stations is ensured by MoUs with the national Hydrometeorological Services and the Central-Asian Institute for Applied Geosciences (CAIAG) for Kyrgyzstan. Members of the Hydrometeorological Services are participating in installations and maintenance trips and technical field and laboratory trainings ensure the knowledge transfer and thus, the sustained operation. Also CAIAG as the regional institutional partner provides expertise in operating the stations. Moreover, all technical components of the stations are commercially available which allows replacing components without GFZ's interaction.

---

## Referee Comment (RC2) · Anonymous Referee #2 · 18 Nov 2020

In this data manuscript, the authors have provided a full description of a hydrometeo-rological dataset from Central Asia and Afghanistan. The dataset has very high quality and is of high interest to the research community for an area with scarce data. The manuscript can be considered for publication after a moderate revision. Below are some comments that can be used to improve the quality of the manuscript.

- The dataset is not easily accessible. I would recommend providing a direct link to the dataset without any need to create a user and password. Data also need a post-processing and the presentation of data must be simple. Use simple indices such as P for precipitation, T for air temperature, RH for relative humidity, and so on for the time

series in the website as well as the tables in the manuscript. Description of the file formats in sections 7.1 and 7.2 is excessive and needs to be polished and simplified.

- Provide plots for monthly climatological averages for the main hydrometeorological variables.

-Are Tipping buckets the only precipitation gauges? Do they only record rainfall (e.g., Figure 4)? How about snowfall? Given the high range of elevation among the stations, a Tipping bucket gauge is not enough for representing snowfall portion of precipitation.

- Spell out all of the acronyms such as GNSS. Spell out all the acronyms on the figures and in the figure captions.

- Combine Figures 2 and 3.
* * *

---

## Referee Comment (RC3) · Anonymous Referee #3 · 19 Nov 2020

This paper presents a detailed description of a station network consisting of 18 stations, which were installed by the German Research Center for Geosciences, Potsdam, Germany and operated today in close collaboration with the Central Asian Institute for Applied Geosciences in Bishkek, Kyrgyzstan. The network is covering several countries in Central Asia. The data of this network is free of charge, real time and online available.

General comments: The paper has a clear structure and explains in detail the installed sensors and the data retrieval. This paper presents a very important dataset generated in a data poor region of the world. Until the 1990s, Central Asia had a very good and

extensive network of hydro-meteorological stations. However, after the collapse of the Sovietunion, most stations in the network were abandoned. Therefore, all efforts to (re-)establish and improve the hydrometeorological station network in all Central Asian countries is very much appreciated, particular the high-mountain stations, because the sensitive cryospheric components such as snow, glaciers and permafrost are currently reacting very sensitive to the atmospheric warming and the development of forecasting tools is only possible when having well validated and calibrated models at hand. These data provide, therefore, a fundamental base for any sound model approaches. In addition, it is a particular asset that the presented data is free of charge, in contrast to the very expensive data when ordering it at the official Hydromet Services.

Specific comments:

- Please provide some more details for the individual sensors used: in general error ranges of the calibrated sensors or e.g. radiation measurement: range of wave length for short wave and long wave radiation sensors.

- Line 188: please delete 'the energy balance between' because the word energy balance should be only used for the total energy balance including all energy fluxes such as the turbulent fluxes and/or ground heat fluxes.

- Line 230: you describe in detail the Snow Pack Analyser but you not describe the above mentioned Cosmic-Ray Neutron Sensing. At which station you operated this system? What are your experiences with this system?

---

## Referee Comment (RC4) · Anonymous Referee #4 · 23 Nov 2020

The high mountains such as the Pamir and Tien Shan where most of the water originates from glaciers and snow provide important water resources for the Central Asia region. However, limited or scarce gauge station measurements lead to a lack of information that are key for water resource managements. To address this issue, the authors provide a dataset including near real-time meteorological and hydrological data collected from 18 remotely operated multi-parameter stations in the high mountains of Central Asia. In this manuscript, the authors provide a detailed description of station infromation, data collection methods as well as data quality issues. Certainly, this dataset is openly accessible, which is of great importance for applications in climate change and water resource management studies. However, there are still additional

work need to be done. Thus, I recommend major revisions before publication. My comments and suggestions are as follows:

1. Firstly, all the figures and tables should be thoroughly revised to make them more readable.

2. The captions of the tables and figures should be more concrete.

3. Although the authors try to show that the stations are located in remote areas and high altitudes, it's hard to get this information from Figures 2 and 3. I suggest the authors provide additional information such as elevation, mean temperature, and mean precipitation on these two Figures.

4. Since precipitation is the most difficult variable to measure, I suggest the authors provide clear figure about compact Weather Transmitter, especially precipitation sensor. Additionally, the authors should provide relevant parameters and measuring error of precipitation sensor.

5. I also suggest the authors provide close-up of Snow Pack Analyzer, which is useful for understanding the description on how to measure snow depth/snow density and its possible error sources.

6. I think it's better to move Parts 7 to supplementary material, since it's not quite important for this manuscript.

7. I suggest the authors add information such as elevation, longitude and latitude of all the stations into Table 2.

8. The authors stated that "the primary purpose of the network is to provide near real-time data for the Hydromet services without major time delay. A consistency or QC on this dataset is beyond the scope of the network operation". However, I think at least the systematic errors and apparent outliers of the data should be removed before the publishing.

9. In lines 49-52, the authors pointed out that "The monitoring network degraded significantly after 1991 mainly due to economic shortening resulting in a lack of information urgently needed for water availability decisions". Do all the stations stop working since 1991? If not, please provide addtional information about which stations are still working. Besides, are the 18 stations enough? How do you optimize the geographic locations of these stations?

10. I suggest publishing a copy of this dataset in the National Tibetan Plateau/Third Pole Environment Data Center http://data.tpdc.ac.cn/en/. This could help facilitating the High Mountain Asia study.

---

## Author Comment (AC2) · 26 Nov 2020

We thank Reviewer #2 for the constructive comments and suggestions. Concerning the four comments, the authors answer as follows:

Comment 1: a) The dataset is not easily accessible. I would recommend providing a direct link to the dataset without any need to create a user and password.

The data can be downloaded at https://kurzelinks.de/romps-data and http://sdss.caiag.kg without any registration or any user/password login which has been already described in the manuscript. With the acceptance of the paper, the

data will be made available at GFZ's central data service long-term archive. The login is only necessary for administrating the web-side.

b) Data also need a postprocessing and the presentation of data must be simple.

As described in chapter 5 of the manuscript, the data has to be considered as raw data which have not undergone any post-processing as this is beyond the scope of the network operation and are subject of each responsible agency or user. Quality control procedures that should be considered are proposed in chapter 8. The Quality Control and dissemination of aggregated values is the responsibility of the national Hydromet-Services. The data can be easily displayed through the SDSS system (sdss.caiag.kg).

c) Use simple indices such as P for precipitation, T for air temperature, RH for relative humidity, and so on for the time series in the website as well as the tables in the manuscript. Description of the file formats in sections 7.1 and 7.2 is excessive and needs to be polished and simplified.

The indices described in the manuscript refer to the technical implemented indices used by the sensors or in the station's datalogger. The data presented here has to be considered as raw data directly coming from the monitoring stations. Changing the indices would require a post-processing of all data. We believe retaining the original abbreviations helps the user in backtracing the data to the original sensors with different accuracies and measurement methodologies.

Section 7.1 & 7.2 are necessary as they describe the storage logic necessary to re-trieve the data.

Comment 2: Are Tipping buckets the only precipitation gauges? Do they only record rainfall (e.g., Figure 4)? How about snowfall? Given the high range of elevation among the stations, a Tipping bucket gauge is not enough for representing snowfall portion of precipitation.

Prior to the network development, we evaluated which sensor types could be safely

used in this remote network. Since then, we haven't changed hardware types to keep the maintenance and operation efforts small. The decision was based on the accuracy of the sensor and the power consumption and maintenance free (or at least minimal) aspects.

Measuring solid precipitation requires a heating system. As the power consumption for the heating is high and the stations depend on solar power only, these systems could not be used for this network.

Comment 3: Spell out all of the acronyms such as GNSS. Spell out all the acronyms on the figures and in the figure captions.

We follow the suggestion of the reviewer and changed the manuscript.

Comment 4: Combine Figures 2 and 3

We take the suggestion of the reviewer and changed the manuscript.

―――――――――――――――――

---

## Author Comment (AC3) · 26 Nov 2020

We thank Reviewer #3 for the constructive comments and suggestions. Concerning the three comments, the authors answer as follows:

Comment 1: Please provide some more details for the individual sensors used: in general error ranges of the calibrated sensors or e.g. radiation measurement: range of wave length for short wave and long wave radiation sensors.

These descriptions and values (e.g. range of wave length) are described in the data format specification for each sensor values which is part of the supplementary material

provided with the manuscript. A detailed description of the sensors itself can be found in the manuals of the sensors. Links to these manuals are listed at the end of the paper's reference list and the manuscript is not overloaded with technical details.

Comment 2: Line 188: please delete 'the energy balance between' because the word energy balance should be only used for the total energy balance including all energy fluxes such as the turbulent fluxes and/or ground heat fluxes.

We agree with the comment and changed the manuscript accordingly.

Comment 3: you describe in detail the Snow Pack Analyser but you not describe the above mentioned Cosmic-Ray Neutron Sensing. At which station you operated this system? What are your experiences with this system?

The CAWa ROMPS stations do not only provide hydrometeorological data, they also serve as a data storage and transmission hub for third party installations. Therefore, other systems such as broadband seismometers, automatic cameras, and a Cosmic-Ray Neutron Sensing sensor are integrated to the ROMPS stations by other users. We refer to the publication of Schattan et al. (doi: 10.1007/s00506-018-0500-x) for further information. CRNS data from Golubin Glacier is distributed through the SDSS (sdss.caiag.kg), but will also be available through GFZ Data Repository.

We have added a table for the 3rd party equipment to the manuscript.

---

## Author Comment (AC4) · 4 Dec 2020

We thank Reviewer #4 for the constructive comments and suggestions. Concerning the ten comments, the authors answer as follows:

Comment 1: Firstly, all the figures and tables should be thoroughly revised to make them more readable.

We carefully revised parts of the figures and tables.

Comment 2: The captions of the tables and figures should be more concrete.

We follow the suggestion of the reviewer and extended the table/figure captions in the

manuscript.

Comment 3: Although the authors try to show that the stations are located in remote areas and high altitudes, it's hard to get this information from Figures 2 and 3. I suggest the authors provide additional information such as elevation, mean temperature, and mean precipitation on these two Figures.

We now included a table with station names and location information including the elevation to the supplementary material.

Comment 4: Since precipitation is the most difficult variable to measure, I suggest the authors provide clear figure about compact Weather Transmitter, especially precipitation sensor. Additionally, the authors should provide relevant parameters and measuring error of precipitation sensor.

Sensor-specific information of the precipitation sensor such as the measuring error and measuring range can be found in the manual of the sensor. We have extended the description of the sensor in the manuscript. Additionally, we have compiled all manuals and datasheets from the sensors to one compressed file to the supplementary material.

Comment 5: I also suggest the authors provide close-up of Snow Pack Analyzer, which is useful for understanding the description on how to measure snow depth/snow density and its possible error sources.

We follow the suggestions of the reviewer and have inserted a close-up picture of the Snow Pack Analyzer.

Comment 6: I think it's better to move Parts 7 to supplementary material, since it's not quite important for this manuscript.

We investigated various publications in this journal and found that the technical description of the data in the manuscript is quite typical for this journal. We have therefore decided not to move it to the supplementary material, as it is necessary to retrieve the data.

[Figure]

Comment 7: I suggest the authors add information such as elevation, longitude and latitude of all the stations into Table 2.

Although this information is part of the individual station documentation that is provided with the supplementary material, we have compiled a table with the locations of the stations and put it to the supplementary material to give a better overview. See also comment 3.

Comment 8: The authors stated that "the primary purpose of the network is to provide near realtime data for the Hydromet services without major time delay. A consistency or QC on this dataset is beyond the scope of the network operation". However, I think at least the systematic errors and apparent outliers of the data should be removed before the publishing.

Due to the symptoms of systematic errors (errors in the configuration or technical errors), these are difficult to detect automatically. Therefore, they can only be eliminated by complex post-processing, which in turn would result in a considerable time-delay. However, this is not the scope of a real-time network operation. When setting up the network, a conscious decision was made to provide near-real-time data without quality control instead of a system that provides quality-controlled data with a time delay of several months.

Comment 9: In lines 49-52, the authors pointed out that "The monitoring network degraded significantly after 1991 mainly due to economic shortening resulting in a lack of information urgently needed for water availability decisions". Do all the stations stop working since 1991? If not, please provide additional information about which stations are still working. Besides, are the 18 stations enough? How do you optimize the geographic locations of these stations?

The monitoring network degraded significantly (Unger-Shayesteh et al., 2015) but did not stop working completely. A more detailed percentage distribution over different years is given in Finaev, 2009 for Tajikistan, in Glazirin, 2009 for Uzbekistan and
Kuzmichenok, 2009 for Kyrgyzstan. Nevertheless, the provision of 18 stations are not enough but was a helpful support to the national Hydromet Services to extend the existing network with a special focus on remote areas with difficult access for local operators. The locations were selected together with the Hydromet Services.

Finaev, A.: Review of hydrometeorological observations in Tajikistan for the period of 1990–2005, Glazirin, G.E.: Hydrometeorological monitoring system in Uzbekistan, Kuzmichenok, V.: Monitoring of water, snow and glacial resources of Kyrgyzstan,

in: Assessment of Snow, Glacier and Water Resources in Asia, edited by: Braun, L. N., Hagg, W., Severskiy, I. V., and Young, G., Selected papers from the Workshop in Almaty, Kazakhstan, 2006, UNESCO-IHP and German IHP/HWRP National Committee, IHP/HWRP-Berichte 8, 55–64, Koblenz, 2009.

Comment 10: I suggest publishing a copy of this dataset in the National Tibetan Plateau/Third Pole Environment Data Center http://data.tpdc.ac.cn/en/. This could help facilitating the High Mountain Asia study.

We have published the data at the GFZ's central data service long-term archive at https://kurzelinks.de/romps-data and the http://sdss.caiag.kg. The SDSS can directly access the data stream from the stations. Creating a copy of the data at a different location carries the risk of inconsistencies.
* * *